# Recent Advances in Molecular Research and Treatment for Melanoma in Asian Populations

**DOI:** 10.3390/ijms26115370

**Published:** 2025-06-03

**Authors:** Soichiro Kado, Mayumi Komine

**Affiliations:** 1Department of Dermatology, Jichi Medical University, 3311-1 Yakushiji, Shimotsuke 329-0498, Tochigi, Japan; m07027sk@jichi.ac.jp; 2Department of Biochemistry, Jichi Medical University, 3311-1 Yakushiji, Shimotsuke 329-0498, Tochigi, Japan

**Keywords:** melanoma, immune checkpoint inhibitors, BRAF-MEK inhibitors, acral and mucosal melanoma, therapeutic resistance

## Abstract

Melanoma treatment comprised a few treatment choices with insufficient efficacy before the emergence of molecularly targeted medication and immune checkpoint inhibitors, which dramatically improved patient outcomes. B-Rapidly Accelerated Fibrosarcoma (BRAF) and Mitogen-Activated Protein Kinase (MAPK) Kinase (MEK) inhibitors significantly improved survival in *BRAF*-mutant melanoma and immune checkpoint inhibitors, such as anti-programmed cell death 1 (PD-1) and Cytotoxic T-Lymphocyte Antigen 4 (CTLA-4) agents, established new standards of care. Challenges remain, however, including the existence of resistance mechanisms and the reduced efficacy of immune-based therapies in Asian populations, particularly for acral and mucosal subtypes. This review highlights historical and current therapeutic advancements, discusses regional considerations, and explores emerging strategies aiming at globally optimizing melanoma management.

## 1. Introduction

Chemotherapy for melanoma before 2011, the year during which vemurafenib and ipilimumab were approved by the FDA (Food and Drug Administration) in the U.S., provided only few choices for treatment, which had insufficient efficacy. At that time, melanoma was recognized as an untreatable malignancy when advanced. In the 1970s, dacarbazine (DTIC) was established as the standard treatment; however, its efficacy was limited, with poor response rates and a minimal impact on overall survival [1]. Subsequently, combination chemotherapy regimens were investigated in an attempt to enhance therapeutic outcomes. Despite these efforts, no survival advantage over single-agent therapies could be demonstrated, leaving melanoma treatment with unmet clinical needs [2].

The therapeutic landscape shifted dramatically with the advent of molecularly targeted therapies and immune checkpoint inhibitors. Targeted agents such as BRAF inhibitors (the first BRAF inhibitor, vemurafenib, was approved by the FDA in 2011 [3]), particularly when combined with MEK inhibitors (the first combination of BRAF and MEK was approved by the FDA for the first time in 2014 (dabrafenib + trametinib) [4]), showed substantial clinical benefit in patients harboring *BRAF* mutations. Concurrently, the introduction of immune checkpoint inhibitors (the first anti-CTLA antibody ipilimumab was approved in 2011 [5]), and notably of anti-PD-1 antibodies (nivolumab [6] and pembrolizumab [7] were approved in 2014 by the FDA), revolutionized the management of advanced melanoma, establishing these agents as cornerstone therapies and setting new standards of care.

In this review, we summarize the historical evolution of melanoma treatments, from conventional chemotherapy to the current era of precision medicine and immunotherapy. Additionally, we discuss emerging therapeutic strategies and potential directions for the future, aiming to address ongoing challenges and optimize outcomes for patients with melanoma. Importantly, we focus on the unique characteristics and clinical features of melanoma in Asian populations, which often differ from those observed in Caucasian patients. By highlighting these distinctions, this review aims to underscore the need for region-specific clinical evidence and to contribute to the development of more tailored treatment strategies for Asian patients.

## 2. Historical Approaches to Melanoma Treatment

### 2.1. Chemotherapeutic Agents

Dacarbazine (DTIC) had been a standard treatment for metastatic melanoma for over 30 years (Figure 1), despite low response rates of 5–12% [8,9,10,11] (Table 1). Adding interferon or tamoxifen to DTIC did not significantly improve outcomes but increased toxicity [12].

Temozolomide has shown no significant differences in response rates or survival outcomes compared to DTIC, with a reported response rate of 13.4% [13,14] (Table 1).

nab-Paclitaxel (nab-PTX, ABI-007, Abraxane) is a 130 nm albumin-bound particle formulation of paclitaxel designed to enhance drug delivery without the need for solvent-based carriers [15]. This solvent-free formulation improves bioavailability and facilitates higher intratumoral drug concentrations, with preclinical studies reporting a 33% increase compared to solvent-based paclitaxel at equivalent doses [16]. Clinical trials have demonstrated its efficacy and safety, establishing its potential advantages in drug delivery and therapeutic outcomes. Notably, a phase III trial of nab-PTX reported a response rate of 15%, with no significant differences in response rate or OS compared to DTIC [17] (Table 1).

No studies have directly compared the efficacy of the carboplatin plus paclitaxel (CBDCA + PTX) regimen with dacarbazine using statistical methods. However, the CBDCA + PTX regimen demonstrated an objective response rate of 18% in a phase III trial for metastatic malignant melanoma [18] (Table 1). Although this was not a direct comparison, it is considered that there may be no significant difference in efficacy between dacarbazine and the CBDCA + PTX regimen.

The CVD regimen is a combination therapy comprising cisplatin, vindesine, and dacarbazine. Phase III clinical trials have reported response rates ranging from 20% to 35% [19] (Table 1). However, despite its moderate response rates, the CVD regimen did not contribute to an improvement in overall survival (OS) [19]. Consequently, it has been reported that the regimen fails to demonstrate significant superiority over dacarbazine monotherapy in clinical outcomes [19].

The regimens mentioned above, including DTIC, temozolomide, nab-PTX, CBDCA+PTX, and CVD therapy, are listed in the latest NCCN Guidelines and remain potential options for chemotherapy in current clinical practice [20]. Although not included in the NCCN Guidelines, DAV therapy has been widely used in Japan. DAV therapy, consisting of dacarbazine (DTIC), nimustine hydrochloride (ACNU), and vincristine (VCR), has been employed as an adjuvant treatment for advanced melanoma for several decades, despite the absence of dedicated clinical trials validating its efficacy [21]. In addition, a modified version, DAV-feron therapy, which includes the incorporation of interferon-beta (IFN-β), has been utilized. These therapeutic approaches are hypothesized to exert anti-melanoma immune effects by modulating tumor-associated macrophages (TAMs) and influencing the tumor microenvironment [21].

Cytotoxic chemotherapy, while its role has markedly diminished with the advent of immune checkpoint inhibitors and targeted therapies, remains a potential therapeutic option in selected clinical scenarios and warrants consideration in tailored treatment strategies.

### 2.2. Treatments Involving the Use of IL-2 or IFN

High-dose interleukin-2 (HD IL-2) monotherapy demonstrated an overall response rate (ORR) of approximately 15–20% in advanced melanoma [22,23], while combination approaches, such as stereotactic body radiotherapy (SBRT), have been reported to improve the ORR to 54% [24] (Table 1). The therapy has the unique ability to induce durable complete responses in a subset of patients, a feature less commonly observed with cytotoxic agents like dacarbazine. However, its clinical adoption remains limited due to significant toxicities, including vascular leak syndrome and systemic inflammatory responses, which require intensive management and restrict its use to specialized centers (Table 1). In Japan, high-dose IL-2 has not been approved for use in the National Health Insurance system.

Interferon-α (IFN-α) therapy has shown limited efficacy. In a randomized trial comparing interferon-α (IFN-α) combined with dacarbazine (DTIC) versus DTIC alone, no significant differences in response rate or overall survival were observed [25,26]. In addition, IFN-α therapy is associated with both hematologic and non-hematologic toxicities, including bone marrow suppression, fever, and myalgia. While these adverse effects are generally manageable and reversible, they often necessitate dose adjustments or treatment delays [14]. Despite its potential immunomodulatory and antiproliferative benefits, the balance between efficacy and tolerability remains a critical consideration in its clinical application [14,27]. Interferon-α has been approved for use as an adjuvant therapy after surgery under the National Health Insurance system in Japan; however, its use has become extremely limited in recent years (Table 1).

By contrast, the local injection of interferon-beta (IFN-β) demonstrated effectiveness in several studies in the literature for cutaneous metastasis and in-transit metastases [28,29,30,31,32]. Several studies have reported the complete or partial remission of melanoma lesions following intralesional IFN-β administration [28,29] (Table 1). Adverse events associated with local IFN-β injection were limited to mild localized swelling, erythema, and inflammation at the injection site [30]. Overall, the treatment demonstrated a favorable tolerability profile. The combination of immune checkpoint inhibitors and interferon-β has shown potential therapeutic benefits [32]. Under the National Health Insurance system in Japan, IFN-β was combined with DAV therapy until recent advancements in molecularly targeted therapies were made. However, clinical evidence supporting its efficacy remains limited, necessitating further investigation through well-designed studies to validate its clinical utility.

## 3. Current Advances in Melanoma Therapy

### 3.1. Molecular Classification of Melanoma (The Cancer Genome Atlas Project)

Genomic profiling has revealed that cutaneous melanoma can be divided into four major molecular subtypes, characterized by their dominant driver mutations. The Cancer Genome Atlas (TCGA) project (2015) classified 333 melanomas into *BRAF*-mutant, *RAS*-mutant, *NF1*-mutant, and Triple Wild-Type (Triple-WT) groups [33]. In this schema, approximately 50–60% of tumors carry activating *BRAF* mutations (predominantly V600E/K), ~20–30% have *RAS* mutations (mostly *NRAS* codon 61), ~10–15% harbor loss-of-function mutations in *NF1*, and the remainder lack alterations in all three (*Triple-WT*) [33]. Notably, the *NF1* subgroup frequently shows a very high mutation burden and *NF1* loss serves as an alternative route to MAPK activation. The Triple-WT group (comprising about 15% of cases) is defined by the absence of hot-spot *BRAF/NRAS/NF1* mutations and is genetically heterogeneous; some Triple-WT tumors instead contain other oncogenic events (e.g., mutations or amplifications in *KIT*, *PDGFRA*, *GNAQ/GNA11*, etc.). In TCGA’s analysis, these four subtypes did not show clear differences in overall survival but did underline the centrality of MAPK-pathway deregulation in melanoma—in fact, >90% of the *BRAF*, *RAS*, and *NF1* subtypes showed UV-signature mutational spectra. Collectively, the TCGA classification highlights that most melanomas harbor mutations converging on the RAS–RAF–MEK–ERK (MAPK) signaling cascade (Figure 2).

### 3.2. Molecularly Targeted Therapies

The development of molecularly targeted therapies has dramatically changed the treatment landscape for malignant melanoma, particularly for cases harboring actionable genetic mutations. These therapies specifically target the molecular drivers of tumor progression, offering superior efficacy compared to traditional cytotoxic agents with less adverse events.

#### 3.2.1. BRAF V600 Mutation

B-Rapidly Accelerated Fibrosarcoma (*BRAF*) mutations, most commonly V600E and V600K, are key drivers of melanoma, occurring in approximately 60% of cutaneous melanoma worldwide [34,35]. These mutations activate the downstream Mitogen-Activated Protein Kinase/Extracellular Signal-Regulated Kinase Signaling Pathway (MEK-MAPK), promoting cell proliferation and survival [35]. Targeted therapies have focused on BRAF inhibitors and their combinations with MEK inhibitors, which have consistently demonstrated enhanced efficacy compared to monotherapy. The combination of dabrafenib and trametinib (D + T) achieves an overall response rate (ORR) of 68%, with a median progression-free survival (PFS) of 11.1 months and a median overall survival (OS) of 25.9 months [36]. In contrast, vemurafenib monotherapy yields lower outcomes, with a PFS of 7.3 months and OS of 16.9 months [37]. Another effective combination, encorafenib and binimetinib (E + B), reported an ORR of 64%, a PFS of 14.9 months, and an OS of 33.6 months in the COLUMBUS trial [38]. Additionally, vemurafenib combined with cobimetinib (V + C) demonstrated an ORR of 70%, a PFS of 12.3 months, and an OS of 22.3 months [39]. The superior efficacy of combination therapies involving BRAF and MEK inhibitors is attributed to their ability to more effectively suppress the MAPK signaling pathway [40]. This dual inhibition strategy not only prolongs PFS and overall survival OS but also mitigates the emergence of resistance mechanisms commonly observed with BRAF inhibitor monotherapy. For example, BRAF inhibitors alone can lead to the compensatory reactivation of the MAPK pathway through upstream receptor tyrosine kinases or secondary *MEK* mutations, which drive continued tumor progression [41]. The addition of MEK inhibitors counteracts these mechanisms by ensuring sustained pathway inhibition, resulting in improved clinical outcomes [42]. Moreover, the safety profile of combination therapy is favorable, as it allows for lower doses of each agent, reducing dose-dependent toxicities while maintaining efficacy [43]. Among the approved combinations, D + T, V + C, and E + B have demonstrated significant improvements in objective response rates (ORRs) and survival metrics compared to monotherapy in *BRAF* V600-mutant melanoma [35,37,38]. However, the therapeutic efficacy of BRAF + MEK inhibitors may vary depending on the specific mutation subtype, such as V600E or V600K [44,45]. Clinically, V600K mutations are more prevalent in older males and are associated with chronic sun damage, which contrasts with the typical characteristics of V600E melanoma [46]. In contrast, V600E mutations are more commonly observed in younger patients and occur predominantly in non-sun-damaged skin [46]. These tumors are often linked to a lower mutational burden compared to V600K and demonstrate a stronger dependency on the ERK pathway, which may explain their relatively higher responsiveness to BRAF+MEK inhibitor therapies [44,45]. These molecular differences are reflected in clinical outcomes, with patients harboring V600K mutations showing lower response rates and shorter progression-free survival compared to those with V600E mutations [45]. For example, the median PFS has been reported as 5.7 months for V600K compared to 7.1 months for V600E [44,45]. This reduced efficacy is likely due to the lower dependency of V600K melanomas on the ERK pathway, limiting the impact of MAPK-targeted therapies [44]. Additionally, V600K melanomas are associated with worse overall prognosis, including higher rates of metastasis to the brain and lungs and shorter intervals from diagnosis to disease progression [45].

Given the higher mutational burden and unique tumor microenvironment of V600K melanomas, these patients may benefit more from immunotherapy compared to those with V600E. Indeed, studies have demonstrated that V600K patients treated with anti-PD-1 therapies exhibit superior outcomes, with longer median progression-free survival (19 months vs. 2.7 months for V600E) and a trend toward improved overall survival (20.4 months vs. 11.7 months for V600E) [44]. These findings suggest that therapeutic strategies for *BRAF*-mutant melanomas should consider the specific mutation subtype to optimize clinical outcomes.

Both *BRAF* V600E and V600K mutations are known to confer sensitivity to molecularly targeted therapies such as BRAF inhibitors. However, resistance to these treatments is a prevalent and significant challenge in clinical practice. Mechanistically, resistance to BRAF inhibitors is largely driven by the reactivation or bypassing of the MAPK signaling pathway. This process is often facilitated by mechanisms such as the upregulation of receptor tyrosine kinases, including platelet-derived growth factor receptor beta (PDGFRβ), or genetic alterations in *NRAS* [47]. These adaptive changes enable tumor cells to evade MAPK pathway inhibition, thereby contributing to therapeutic resistance. Studies indicate that resistance emerges in a substantial proportion of patients, with estimates suggesting that up to 50% of cases experience resistance within 6 to 12 months following the initiation of therapy [48,49].

Efforts to mitigate resistance have led to the exploration of several therapeutic strategies. Combination therapy with BRAF inhibitors and MEK inhibitors has shown promise in delaying the onset of resistance by simultaneously targeting multiple nodes within the MAPK pathway [43,48,49]. Additionally, the integration of immune checkpoint inhibitors (ICIs), such as anti-PD-1 and anti-CTLA-4 antibodies, into treatment regimens has emerged as a compelling approach. This combination leverages the synergistic potential of immune activation alongside molecularly targeted therapy, potentially enhancing treatment efficacy and providing a durable response [50,51]. While the combination of immune checkpoint inhibitors with BRAF and MEK inhibitors has emerged as a promising therapeutic strategy for patients with *BRAF*-mutant melanoma, alternative approaches to overcoming resistance are currently under investigation. Among these, the inhibition of the glucocorticoid receptor (GCR) has been explored as a potential means to counteract resistance by modulating tumor cell survival pathways and mitigating the effects of stress hormone signaling. Estrela et al. demonstrated that GCR antagonism overcomes resistance to BRAF inhibition in *BRAF* V600E-mutated metastatic melanoma through the modulation of stress response pathways [52]. Additionally, fibroblast growth factor 1 (FGF1) inhibitors have been studied in this context, as the upregulation of FGF1 has been implicated in resistance to BRAF and MEK inhibition, and its blockade may help to resensitize melanoma cells to targeted therapy. Wang et al. reported that the activation of the FGFR cascade leads to sustained ERK signaling, and adding FGF1 inhibitors to BRAF/MEK inhibitors may restore treatment sensitivity [53]. Moreover, the inhibition of p90 ribosomal S6 kinase (RSK) has demonstrated potential in suppressing protein synthesis and tumor proliferation in melanoma cells that have developed dual resistance to BRAF and MEK inhibitors. BI-D1870 and BRD7389, two RSK inhibitors, were found to significantly reduce tumor growth and protein synthesis in melanoma cell lines exhibiting dual resistance [54].

Furthermore, histone deacetylase (HDAC) inhibitors have gained attention due to their ability to downregulate the microphthalmia-associated transcription factor (MITF), which plays a crucial role in melanoma progression and therapeutic resistance. Aida et al. demonstrated that MITF suppression via HDAC inhibition enhances the apoptotic response of melanoma cells to BRAF inhibitors in preclinical models [55].

These alternative strategies hold promise for overcoming acquired resistance to BRAF and MEK inhibitors, but further research and clinical validation are warranted to establish their efficacy and safety in clinical settings.

#### 3.2.2. KIT Mutation

*KIT* mutations occur in approximately 9.5% of melanomas, with notable variation among subtypes [56]. Acral melanomas exhibit *KIT* mutations in 8.3–23% of cases, while mucosal melanomas have the highest reported frequency, ranging from 15.6% to 50% [57,58]. Unlike *BRAF* mutations, *KIT* mutations are rare in cutaneous melanoma, with a reported frequency ranging from 1.7% to 4.3% [57,58].

These mutations, primarily affecting exons 11 and 13, lead to constitutive activation of tyrosine kinase signaling pathways, contributing to melanoma progression [57]. Among these, mutations in exon 11 (e.g., L576P) and exon 13 (e.g., K642E) are the most prevalent, accounting for approximately 40–55% of mutations, and are associated with the activation of MAPK and PI3K/AKT pathways [57,58].

Clinical studies demonstrate an ORR to KIT inhibitors, such as imatinib, nilotinib, and dasatinib, ranging from 15 to 40%, with the highest efficacy observed in cases with specific mutations like L576P and K642E [59,60]. For example, imatinib achieved an ORR of 24.4% and a disease control rate (DCR) of 66.7% in patients with exon 11 or 13 mutations [61]. Nilotinib showed comparable outcomes, with an ORR of 26% and DCR of 73.8%, particularly in patients with the aforementioned mutations [61]. However, data on the efficacy of KIT inhibitors in patients with mutations outside these exons are limited [59]. Furthermore, in the absence of *KIT* mutations but in the presence of gene amplification, the therapeutic efficacy of these agents is considerably limited [59,62].

Despite these advancements, resistance to KIT inhibitors remains a significant challenge, often driven by secondary mutations, activation of alternative pathways, and microenvironmental factors. Mechanisms of resistance include the reactivation of downstream signaling pathways, such as MAPK and PI3K/AKT, and contributions from tumor-associated macrophages and angiogenic factors [61]. To counter these, combinatorial therapeutic approaches targeting multiple pathways have been explored. Preclinical studies have shown that the dual inhibition of PI3K and MAPK pathways enhances antitumor efficacy [57,61]. Furthermore, novel agents like ponatinib, approved by the FDA for CML and Ph + ALL but not yet for melanoma, which targets KIT and angiogenic pathways, have demonstrated potential in overcoming resistance [61]. Preclinical studies of ponatinib reported promising results, demonstrating potent antitumor effects in *KIT*-mutant melanoma models [63].

#### 3.2.3. NRAS Mutations

*Neuroblastoma RAS viral oncogene homolog (NRAS)* mutations occur in 15–20% of melanomas and are associated with aggressive behavior and poor prognosis [64]. These mutations activate multiple signaling pathways, including PI3K/Akt and MEK-ERK, promoting cancer growth [65]. While targeting *NRAS* directly has been ineffective, inhibiting downstream effectors or KRAS shows promising results [66]. MEK inhibitors, particularly binimetinib, have demonstrated activity in *NRAS*-mutant melanoma [64,67]. Combining MEK inhibitors with CDK4/6 inhibitors appears to be a promising strategy [64,68]; however, it has not yet been approved by the FDA for melanoma. Other potential combinations include MEK inhibitors with FAK inhibitors, autophagy inhibitors, or pan-RAF inhibitors [69]; however, these treatments have not been approved for melanoma treatment by the FDA so far. The FAK inhibitor defactinib, together with avutometinib (an MEK/RAF inhibitor), was put forward for FDA approval in 2025 for ovarian cancer. Tovorafenib, which inhibits multiple RAF family kinases, was approved for pediatric low-grade glioma in 2024. Immunotherapies, especially immune checkpoint inhibitors, may be particularly effective in *NRAS*-mutant melanoma [66]. Despite these advances, the ideal treatment for *NRAS*-mutant melanoma remains elusive, necessitating further research [70,71].

#### 3.2.4. ROS1 Fusions and Mutations

ROS1 fusions have been identified across various cancer types, with particular clinical relevance in non-small cell lung cancer (NSCLC) [72,73], where targeted therapies such as crizotinib [74] and entrectinib [75] have demonstrated significant efficacy. In contrast, the pathogenic significance of ROS1 mutations in NSCLC remains unclear, as their role in tumorigenesis and response to therapy has yet to be well-defined [76].

In melanoma, both *ROS1* mutations and *ROS1* fusions have been reported, though their prevalence and implications differ. *ROS1* mutations are more frequently observed, occurring in approximately 14.8% to 25.0% of cases [76], while *ROS1* fusions are rare and detected in only around 1% of melanomas [77]. Emerging evidence suggests that melanomas harboring *ROS1* mutations may exhibit heightened sensitivity to immune checkpoint inhibitors (ICIs), including PD-1 and CTLA-4 inhibitors [76]. This is thought to be associated with a significantly higher tumor mutational burden (TMB) in *ROS1*-mutated melanomas, potentially enhancing neoantigen presentation and immune system activation.

On the other hand, *ROS1* fusions in melanoma may represent a therapeutic target akin to NSCLC. A case of melanoma harboring *GOPC-ROS1* fusion has been reported, where treatment with crizotinib resulted in a favorable clinical response [78]. Similarly, entrectinib has shown potential efficacy in *ROS1* fusion-positive melanomas [77], suggesting that *ROS1* fusion-directed therapies could be a viable treatment strategy in select cases.

Despite these promising findings, further investigation is warranted to elucidate the precise role of *ROS1* mutations and fusions in melanoma. Future studies should focus on clarifying their oncogenic mechanisms, therapeutic implications, and potential as predictive biomarkers for both immunotherapy and targeted therapy.

#### 3.2.5. NTRK Fusions and Mutations

Similarly to *ROS1* gene fusions, *Neutrophic tropomyosin receptor kinase* (*NTRK*) alterations in melanoma encompass both gene fusions and somatic mutations. While *NTRK* gene fusions are rare, they are recognized as oncogenic drivers across multiple cancer types, including melanoma [79,80]. The prevalence of *NTRK* gene fusions varies among melanoma subtypes, being notably higher in spitzoid melanoma (21–29%) compared to other subtypes, where the frequency is reported to be < 1–2.5% [79]. Identified *NTRK* fusion partners in melanoma include *TRIM63-NTRK1*, *DDR2-NTRK1*, *GON4L-NTRK1*, *TP53-NTRK1*, *LMNA-NTRK1*, *TRAF2-NTRK2*, *ETV6-NTRK3*, *MYO5A-NTRK3*, and *MYH9-NTRK3*.

In contrast, somatic *NTRK* mutations appear to be more common than gene fusions in cutaneous melanoma, with a reported frequency of 19.5% (86/440 cases) [81]. Notably, patients with *NTRK* mutations have demonstrated a significantly higher ORR to immune checkpoint inhibitors (ICIs) compared to those without *NTRK* mutations (42.8% vs. 23.5%, *p* = 0.002) [81]. These findings suggest a potential role of *NTRK* mutations as predictive biomarkers for ICI therapy response in melanoma.

Given the therapeutic implications of *NTRK* fusions, molecular screening for these alterations is recommended in *BRAF*, *NRAS*, and *KIT* wild-type melanomas [80]. Targeted therapies with TRK inhibitors, such as larotrectinib and entrectinib, have shown high response rates (>75%) in *NTRK* fusion-positive cancers across different histologies [82,83]. These inhibitors are generally well-tolerated, with on-target adverse events being rare [82]. However, acquired resistance through *NTRK* kinase domain mutations has been reported, which may be addressed by second-generation TRK inhibitors such as selitrectinib (LOXO-195) and repotrectinib (TPX-0005) [82].

The FDA granted accelerated approval to larotrectinib in 2018 and to entrectinib in 2019 for the treatment of solid tumors harboring *NTRK* gene fusions, with entrectinib also receiving approval for pediatric patients. Clinical trials have suggested the efficacy of TRK inhibitors in melanoma, although further studies are warranted to establish their role in this malignancy [84,85].

### 3.3. Immunotherapy

Immune checkpoint inhibitors (ICIs) represent a cornerstone in the management of unresectable and metastatic melanoma, with two primary classes initially approved for clinical use: CTLA-4 inhibitors and PD-1 inhibitors. The CTLA-4 inhibitor, ipilimumab, enhances T cell activity by disrupting the immunosuppressive interaction between CTLA-4 and B7 [86,87,88]. Since its approval by the FDA in 2011 and in Japan in 2015, ipilimumab has been associated with prolonged overall survival (OS) [5]. A phase III trial (CA184-024) demonstrated that ipilimumab combined with dacarbazine improved OS compared to dacarbazine alone [89], although this combination was limited by an increased risk of hepatotoxicity [90]. Consequently, ipilimumab monotherapy or its combination with PD-1 inhibitors has become the preferred strategy in advanced melanoma.

PD-1 inhibitors, such as pembrolizumab and nivolumab, function by blocking the PD-1 receptor on T cells, thereby enhancing the immune system’s capacity to recognize and eliminate cancer cells [91,92]. Nivolumab was approved by the FDA and by Japanese authorities in 2014, while pembrolizumab was approved by the FDA in 2014 and for use in Japan in 2016. Pembrolizumab has demonstrated superior efficacy over ipilimumab in both treatment-naive patients with metastatic or unresectable melanoma (KEYNOTE-006 trial [93,94]) and in previously treated patients with metastatic or unresectable disease following chemotherapy (KEYNOTE-002 [95]). Similarly, nivolumab has demonstrated significant efficacy as monotherapy and in combination with ipilimumab, as shown in the CHECKMATE-037 trial [6], which compared nivolumab to chemotherapy, the CHECKMATE-066 trial [96], which evaluated nivolumab against dacarbazine, and the CHECKMATE-067 trial [97,98], which evaluated the nivolumab and ipilimumab combination against nivolumab and ipilimumab. Data from these pivotal clinical trials underscore the efficacy of ICIs in improving clinical outcomes in melanoma patients. These findings support the use of pembrolizumab and nivolumab as key agents in the therapeutic arsenal for unresectable melanoma. It remains unclear precisely which patients with melanoma benefit from ICIs. PD-1 inhibitors, such as pembrolizumab and nivolumab, exert therapeutic effects primarily by blocking the interaction between the PD-1 receptor on T cells and its ligand, PD-L1, expressed on tumor cells. Initially, tumor PD-L1 expression was anticipated to correlate positively with clinical response. However, the clinical implementation of PD-L1 as a reliable biomarker has been challenging due to its dynamic variability and intratumoral heterogeneity, as well as the observation that some patients with PD-L1-negative tumors also respond favorably to PD-1 inhibitors [99].

Recent studies have demonstrated that PD-1, traditionally considered exclusive to immune cells, is also expressed on melanoma cells [100,101]. In melanoma, the binding of PD-1 onto tumor cells with PD-L1 expressed by the surrounding tumor or stromal cells can activate the mTOR signaling pathway, thereby directly promoting tumor growth independently of immune cell interactions [101]. Notably, PD-1 inhibitors may exert their therapeutic efficacy not only through immune modulation but also by directly interacting with PD-1 expressed on melanoma cells [101]. Furthermore, recent evidence indicates that androgen receptor activation in melanoma cells may interfere with the efficacy of ICIs [100]. Detailed biomarkers for treatment response will be discussed in a separate section; however, numerous studies are currently underway to identify reliable biomarkers predictive of response to ICIs.

Lymphocyte-activation gene 3 (LAG-3) inhibitors represent the third class of immune checkpoint inhibitors (ICIs) and have been approved by the FDA for the treatment of melanoma in combination with nivolumab since 2022.

LAG-3 is an inhibitory molecule expressed on the surface of T cells, where it negatively regulates T cell activation. Its blockade enhances immune responses by facilitating T cell proliferation and function [102]. Notably, LAG-3 interacts with major histocompatibility complex (MHC) class II molecules, further modulating immune activation [103,104].

The phase II/III RELATIVITY-047 trial demonstrated that the combination of nivolumab and relatlimab, an anti-LAG-3 monoclonal antibody, significantly prolonged progression-free survival (PFS) compared to nivolumab monotherapy in patients with advanced melanoma (median PFS: 10.1 months vs. 4.6 months, respectively) [105]. While this combination has been incorporated into clinical practice in the United States, it has not yet been approved in Japan. Nevertheless, it is listed in the National Comprehensive Cancer Network (NCCN) guidelines, indicating its potential for future adoption in real-world clinical settings.

Clinical trials investigating the immune checkpoint inhibitors discussed above are summarized in Table 2.

### 3.4. Talimogene Laherparepvec (T-VEC)

Talimogene laherparepvec (T-VEC) is an oncolytic virus therapy derived from the herpes simplex virus type 1 (HSV-1) that has been genetically engineered to selectively replicate within tumor cells and enhance systemic anti-tumor immunity. T-VEC is administered via intralesional injection and functions by inducing direct tumor cell lysis and promoting an immunogenic response, which includes the upregulation of the granulocyte-macrophage colony-stimulating factor (GM-CSF) to recruit antigen-presenting cells and stimulate cytotoxic T lymphocytes [106].

A pivotal phase III trial (OPTiM) compared T-VEC with recombinant GM-CSF in patients with unresectable stage IIIB-IV melanoma [5]. The study demonstrated that T-VEC significantly improved the durable response rate (DRR) compared to GM-CSF (16.3% vs. 2.1%, *p* < 0.001) and exhibited greater efficacy in patients with earlier-stage disease (stage IIIB-IIIC: DRR 33.0%) [5]. Furthermore, the therapy elicited systemic effects, leading to tumor regression in non-injected lesions, indicative of an abscopal immune response [5,107].

In efforts to enhance therapeutic outcomes, a phase II study evaluated the combination of T-VEC with ipilimumab, a CTLA-4 checkpoint inhibitor, in unresectable melanoma [108]. The combination demonstrated superior overall response rates (ORR: 39% vs. 18%, *p* = 0.002) compared to ipilimumab monotherapy [108]. Despite the promising response rate, progression-free survival (PFS) and overall survival (OS) benefits remain under investigation.

T-VEC is primarily recommended for patients with stage IIIB-IV melanoma with injectable cutaneous, subcutaneous, or nodal metastases. However, its efficacy diminishes in stage IV-M1b/M1c disease, where systemic immunotherapy or targeted therapy is often preferred [20]. Additionally, data suggest that T-VEC is most effective in treatment-naïve patients, as prior exposure to systemic therapies may attenuate the immunogenic response [109]. This may be due to the immunosuppressive alterations in the tumor microenvironment following previous systemic interventions [107].

The superior efficacy of T-VEC in initial treatment settings may be attributed to its mechanism of action, which involves both direct viral-mediated oncolysis and the stimulation of a systemic immune response. In patients previously treated with immunosuppressive agents, the tumor microenvironment may develop resistance mechanisms that limit T-VEC’s immune-stimulatory effects [107].

T-VEC represents a novel approach to melanoma treatment by leveraging local oncolysis and systemic immune activation. While its clinical utility is predominantly within early metastatic disease, ongoing studies are exploring its integration with immune checkpoint inhibitors to optimize therapeutic efficacy in advanced melanoma cases.

## 4. Specific Considerations for Melanoma Treatment in Asian Populations

### 4.1. Immune Checkpoint Inhibitor Therapy in Asian Populations

Acral lentiginous melanoma (ALM) and mucosal melanoma (MM) are predominant melanoma subtypes in Asian populations, contrasting with the higher prevalence of cutaneous melanoma (CM) in Caucasians. ALM accounts for approximately 40–71% of melanomas in Asian populations [110,111,112], and MM constitutes about 15–27% [110,113,114]. In contrast, ALM is relatively rare among Caucasians, comprising only 1–3% of melanomas [114,115], whereas MM accounts for approximately 1% [116].

These subtypes are biologically distinct, exhibiting a lower tumor mutational burden (TMB) and different oncogenic drivers compared to CM [117].

The efficacy of immune checkpoint inhibitors (ICIs) is notably reduced in ALM and MM. While the objective response rates (ORRs) for anti-PD-1 therapy in CM range between 30% and 40% [6,93,96,118,119], the ORRs for ALM and MM are significantly lower, ranging from 14% to 19% [117]. The reduced tumor mutational burden (TMB) in ALM and MM, attributable to their occurrence in UV-shielded regions, is a primary factor contributing to the diminished efficacy of ICIs [120,121,122].

Even for CM, recent studies suggest that Asian patients may experience lower ICI efficacy than Caucasians. This observation is attributed to lower TMB in Asian populations. For instance, Huang et al. noted that Asian cutaneous melanoma (CM) exhibits a median TMB of 5.1 mutations per megabase (mut/Mb), which is significantly lower than that observed in Caucasian populations [123]. Similarly, Hida et al. reported that the mean TMB of Asian CM is 4.6 mut/Mb, suggesting that the TMB range for Asian CM can be approximated as 4.6–5.1 mut/Mb [124]. In contrast, Caucasian CM is characterized by a markedly higher mean TMB of 49.17 mut/Mb [125], highlighting a significant disparity in the mutational landscape between these populations.

These findings suggest that the efficacy of ICIs may vary among different ethnic groups, even within the same subtype of malignant melanoma. To address this potential variability, prospective studies evaluating the efficacy of ICIs across diverse ethnic populations may be warranted in the future.

### 4.2. Monotherapy vs. Combination Therapy for Asian Melanoma Patients

The question of whether combination therapy of different-class ICIs offers superior benefits compared to monotherapy in treating ALM and MM remains a crucial concern. Retrospective studies and clinical trials have produced mixed results. For ALM, combination therapy has demonstrated higher ORR in nail apparatus melanoma (e.g., 61% with nivolumab plus ipilimumab vs. 10% with monotherapy) but has shown no significant survival advantage for lesions on other acral sites, such as palms and soles [126]. Similarly, studies on MM have reported improved ORRs with combination therapy compared to monotherapy (43% vs. 26%, respectively), but recent analyses indicate no statistically significant differences in overall survival (OS) or progression-free survival (PFS) between the two approaches [127].

Given these findings, the routine use of combination therapy for Asian patients with ALM or MM may not be warranted in all cases. While combination therapy may be considered for selected subtypes, such as nail apparatus ALM, or in cases of monotherapy failure, its benefits in broader contexts appear limited. Importantly, clinical decisions should be guided by individual patient characteristics, including anatomical site, molecular features, and performance status, to maximize therapeutic outcomes. Further prospective studies are essential to clarify the optimal therapeutic strategies for these challenging melanoma subtypes.

### 4.3. Therapeutic Approaches for BRAF-Positive Cases in Asian Populations

Recent studies, including the B-CHECK-RWD study conducted in Japan, indicate significant differences in the efficacy of immune checkpoint inhibitors (ICIs) between Asian and Caucasian populations with advanced *BRAF* V600-mutant melanoma. While ICIs such as nivolumab and the combination of nivolumab plus ipilimumab demonstrate promising outcomes in Western populations, their efficacy appears comparatively reduced in Asian cohorts [128].

In contrast, BRAF/MEK inhibitors maintain comparable efficacy between Asian and Caucasian populations. Data show ORRs of 68% and 67% in Caucasian and Japanese cohorts, respectively, when treated with dabrafenib plus trametinib [128]. Despite this equivalence, the sequencing of BRAF/MEK inhibitors and ICIs remains a pivotal consideration. The DREAMseq and SECOMBIT trials in Western populations advocate for upfront ICI therapy due to its potential for durable survival benefits [129,130]. However, the suboptimal response of ICIs in Asian patients complicates the direct adoption of this strategy.

The B-CHECK-RWD study highlights that salvage therapy with ICIs following disease progression on BRAF/MEK inhibitors yields diminished outcomes, whereas BRAF/MEK inhibitors retain efficacy irrespective of treatment line [128]. Consequently, the choice of first-line therapy in Asian patients necessitates a case-by-case approach, balancing disease characteristics, patient tolerance, and access to therapies.

## 5. Future Perspectives on Melanoma Management

Recent advancement in immunology, molecular biology and tumor immunology enabled us several possible ways to treat melanoma.

First, as aforementioned, to target potential pro-proliferating signaling molecules, such as *BRAF*, *KIT*, *NRAS*, *TERT*, and *CCND1*, most of which harbor constitutive active mutations in these molecules, treatments should target signaling pathways leading to tumor cell proliferation [123]. *TERT* promoter mutations, which promote telomerase activity and enable the indefinite replication of tumor cells, are particularly common in acral melanomas and represent potential therapeutic targets [117,131]. Amplifications in *CCND1*, which drive uncontrolled cell cycle progression via CDK4/6 activation, are another hallmark of acral melanoma, highlighting a distinct pathway compared to UV-induced melanomas [123].

The second way to combat melanoma is to control the tumor microenvironment, with tumor-infiltrating lymphocytes (TILs) playing a key role in this approach. Lifileucel, an autologous TIL therapy, has shown efficacy even in patients who have failed immune checkpoint inhibitors (ICIs) and BRAF/MEK-targeted therapies, as demonstrated in the Phase II C-144-01 trial, which reported an objective response rate (ORR) of 31.4% [132]. Having received FDA approval in the United States, lifileucel is now included in the NCCN guidelines as a promising treatment strategy [20], though it remains unapproved in Japan. The treatment involves the surgical resection of tumor tissue, followed by a 22-day ex vivo expansion of TILs, lymphodepleting chemotherapy with cyclophosphamide and fludarabine, and the subsequent administration of high-dose IL-2 to support TIL persistence. However, significant challenges remain, including the complexity and cost of treatment, the risk of severe adverse events such as thrombocytopenia, capillary leak syndrome, and hypotension, and the need for specialized centers to administer therapy [132]. Despite these limitations, TIL therapy represents a valuable addition to the evolving landscape of melanoma treatment.

The third method is chimeric antigen receptor T (CAR-T) cell therapy, an innovative immunotherapeutic strategy that has demonstrated remarkable efficacy in hematologic malignancies, leading to its FDA approval for specific leukemia and lymphoma subtypes [133,134,135,136]. While CAR-T therapy for melanoma is not yet widely established, ongoing clinical investigations suggest its potential in targeting melanoma-specific antigens [137].

One promising target in melanoma is tyrosinase-related protein 1 (TYRP1), which is highly expressed in tumor cells but has limited expression in normal tissues [138,139]. Preclinical studies have provided compelling evidence of the efficacy and safety of TYRP1-directed CAR-T cells, showing robust antitumor activity in both in vitro and in vivo models [140].

Although the application of CAR-T therapy in solid tumors remains in its early stages, ongoing clinical trials and advancements in antigen selection, CAR engineering, and immunosuppressive microenvironment modulation hold promise for expanding its therapeutic potential in melanoma.

A fourth strategy involves combining immune checkpoint inhibitors (ICIs) with other therapeutic modalities to enhance antitumor efficacy and overcome resistance (Table 3). While ICIs have significantly improved outcomes in melanoma, intrinsic and acquired resistance remains a challenge [141]. To address this, a number of combination strategies have been developed to target complementary immune-inhibitory pathways.

Among these, LAG-3 has gained particular attention. LAG-3 is an inhibitory receptor co-expressed with PD-1 on exhausted T cells, and its blockade has been shown to reinvigorate antitumor immunity. The combination of the anti-PD-1 antibody nivolumab with the anti-LAG-3 antibody relatlimab demonstrated improved efficacy compared to anti-PD-1 monotherapy, leading to an FDA approval of the combination regimen in 2022 [105]. This dual-checkpoint blockade represents a rational approach to enhance immune activation by targeting synergistic inhibitory signals.

Radiotherapy and certain chemotherapeutic agents can induce immunogenic cell death (ICD), enhancing tumor antigen presentation and promoting anti-tumor immunity [141,142]. Clinical studies suggest that concurrent or sequential radiotherapy with ICIs may augment systemic responses, particularly in patients with brain metastases [143,144]. Recent clinical trials are investigating the combination of ICIs with radiation therapy or cytotoxic agents [105,145,146,147,148,149,150,151,152,153].

Oncolytic viruses represent another promising strategy. T-VEC, an intralesional herpesvirus expressing GM-CSF, has shown synergistic effects with ICIs in early-phase studies [154]. However, a phase III trial (MASTERKEY-265) of T-VEC combined with pembrolizumab did not demonstrate superiority over pembrolizumab alone, highlighting the importance of patient selection [155]. Novel oncolytic agents, such as RP1, are being tested in anti-PD-1-refractory melanoma and have demonstrated encouraging response rates (IGNYTE-3) [156].

Therapeutic cancer vaccines, particularly personalized neoantigen vaccines, are emerging as valuable partners for ICIs. In the phase II KEYNOTE-942 trial, the mRNA-4157 vaccine combined with pembrolizumab significantly improved recurrence-free survival in patients with resected high-risk melanoma [157]. Additional studies, including KEYNOTE-D36, are exploring peptide- and dendritic cell-based vaccines in advanced settings [158].

Adoptive cell therapy, particularly tumor-infiltrating lymphocyte (TIL) therapy, is also being investigated in combination with ICIs to improve treatment durability. TIL therapy involves harvesting and expanding autologous T cells from resected tumors, then reinfusing them into the patient following lymphodepletion. This approach has shown promising results in heavily pretreated melanoma patients. In the ongoing phase III TILVANCE-301 trial, the combination of lifileucel (an autologous TIL product) with pembrolizumab is being compared to pembrolizumab monotherapy in treatment-naïve patients with advanced melanoma [159]. This study aims to determine whether the addition of TIL therapy can enhance the efficacy of first-line immunotherapy. Early-phase data have already suggested durable responses, particularly in patients with prior resistance to checkpoint blockade.

Collectively, these combination approaches aim to overcome immune resistance by promoting T cell priming, antigen release, and favorable remodeling of the tumor microenvironment, and they are supported by a growing body of preclinical and clinical evidence.

**Table 3 ijms-26-05370-t003:** Summary of clinical trials evaluating combination strategies with immune checkpoint inhibitors in melanoma.

Phase	Trial Name	Intervention vs. Control	Patient Population	Reference
II/III	RELATIVITY-047	Nivolumab + Relatlimab vs. Nivolumab	Treatment-naive advanced melanoma	[105,145]
II	RadVax	Nivolumab + Ipilimumab + HFRT vs. Nivolumab + Ipilimumab alone	Unresectable stage IV melanoma	[146]
I	NCT02858869	Pembrolizumab + SRS (single arm)	Melanoma or NSCLC with untreated brain metastases	[147]
I	NCT02716948	Nivolumab + SRS (single arm)	Melanoma or NSCLC with untreated brain and/or spine metastases	[148]
II	NIRVANA	Nivolumab + multisite HFRT (single arm)	Metastatic melanoma; no prior systemic therapy	[150]
II	CHEERS	Standard-of-care ICI therapy combined with SBRT vs. standard-of-care ICI therapy	Advanced or metastatic melanoma	[152,153]
II	NCT02617849	Pembrolizumab in combination with carboplatin/paclitaxel (single arm)	Metastatic malignant melanoma	[153]
I/II	IGNYTE-3	Nivolumab + RP1 oncolytic HSV-1 or Rp1 alone	Advanced solid tumor (including melanoma) refractory to anti-PD-1 therapy	[156]
II	KEYNOTE-D36	Pembrolizumab + EVX-01 vs. historical control (no concurrent control arm)	Treatment-naive advanced melanoma	[158]
III	TILVANCE-301	Pembrolizumab + Lifileucel TIL vs. Pembrolizumab alone	Treatment-naive advanced melanoma	[159]
II	ABC-X	Nivolumab + Ipilimumab + SRS vs. Nivolumab + Ipilimumab alone	Melanoma brain metastases; treatment-naïve	[160]
III	MASTERKEY-265	Pembrolizumab + T-VEC vs. Pembrolizumab + placebo	Unresectable stage IIIB–IVM1c melanoma, anti-PD-1 naïve, injectable lesions required	[161]
II	KEYNOTE-942	mRNA-4157 (V940) + Pembrolizumab vs. Pembrolizumab alone	Resected high-risk cutaneous melanoma	[162]

Note: HFRT: hypofractionated radiotherapy; SRS: stereotactic radio surgery; SBRT: stereotactic body radiotherapy; ICI: immune checkpoint inhibitors.

The fifth emerging avenue in melanoma immunotherapy focuses on natural killer (NK) cells, which are increasingly recognized as key effectors in antitumor immunity [163,164]. Recent evidence highlights the role of androgen receptor (AR) signaling in impairing NK cell-mediated cytotoxicity [100]. In melanoma cells, ligand-activated AR promotes the shedding of the NKG2D ligand MICA via the upregulation of ADAM10, a disintegrin and metalloprotease [100]. AR forms a functional complex with β1 integrin and ADAM10, leading to the loss of surface MICA and reduced NK cell recognition. This mechanism facilitates immune evasion, and high serum levels of soluble MICA correlate with a poor response to anti-PD-1 therapy [100]. Sex-based disparities in NK cell function may further compound this immune escape. Females generally exhibit more robust NK activity, while androgens in males may dampen innate immune responses, contributing to worse melanoma outcomes [165,166,167,168,169]. Consequently, the therapeutic blockade of AR represents a compelling strategy. Preclinical models suggest that AR antagonists, such as bicalutamide or enzalutamide, can restore NK-mediated cytotoxicity and synergize with immune checkpoint inhibitors [100]. These findings warrant the future clinical evaluation of AR-targeted therapies in melanoma, particularly in male patients or those with high AR expression.

As a seventh emerging strategy, the identification and validation of biomarkers closely associated with therapeutic response and prognosis have become critical components of melanoma management. Recent studies have identified multiple molecular biomarkers in melanoma associated with therapeutic response and prognosis in the context of immune checkpoint inhibitors (ICIs). PD-L1 expression, while a logical predictor for anti-PD-1/PD-L1 therapies, has shown limited utility in melanoma [170]. Many patients with PD-L1-negative tumors still respond to anti-PD-1 therapy [171], underscoring the need for better biomarkers. High tumor mutational burden (TMB), a hallmark of UV-induced melanoma, is associated with a greater likelihood of response to ICIs [172]. The IFN-γ signature, characterized by an elevated expression of interferon-γ-responsive genes such as CXCL9, CXCL10, IDO1, *IFNG*, and HLA-DR within the tumor microenvironment, reflects an inflamed immune state enriched with activated T cells. This gene expression profile strongly predicts therapeutic responsiveness to anti-PD-1 therapy [173]. Conversely, tumors with low IFN-γ signature (non-inflamed) rarely respond to single-agent immunotherapy; these “cold” tumors might require combination strategies (e.g., adding CTLA-4 blockade, or novel agents to induce T cell infiltration). Overall, the T cell-inflamed GEP is a robust biomarker to predict benefit from PD-1/L1 inhibitors [173]. Tumor-Infiltrating Lymphocytes (TILs) and immune cells represent essential biomarkers reflecting the tumor immune environment. High densities of CD8^+^ cytotoxic T cells within tumors correlate with better responsiveness to ICIs, particularly in the presence of PD-L1 expression [174,175]. A higher CD8:FOXP3 T cell ratio, indicative of increased effector T cells relative to regulatory T cells, also predicts favorable responses [176]. Conversely, immunosuppressive cell populations, such as PD-L1⁻ M2-polarized macrophages, myeloid-derived suppressor cells (MDSCs) [176], and elevated inflammatory markers (IL-6, CRP, neutrophil-to-lymphocyte ratio) [177] are associated with a poor prognosis and resistance to ICIs. Indoleamine 2,3-dioxygenase (IDO1), an enzyme that contributes to immune suppression by degrading tryptophan, is typically associated with a poor prognosis; however, paradoxically, elevated IDO1 expression in tumors might indicate a favorable response to CTLA-4 blockade, as it suggests a regulatory mechanism that checkpoint inhibitors can overcome [173].

Emerging Immune Targets such as LAG-3 are increasingly recognized biomarkers. LAG-3, commonly co-expressed with PD-1 on exhausted T cells, has shown potential as both a predictive biomarker and therapeutic target. The combination of anti-LAG-3 (relatlimab) and nivolumab recently demonstrated clinical efficacy in advanced melanoma [105].

Host Factors, notably specific HLA genotypes and the gut microbiome, significantly impact immunotherapy outcomes. Certain HLA class I alleles (e.g., HLA-B44 supertype, HLA heterozygosity) correlate with improved ICI responses [178]. Furthermore, the gut microbiome composition, particularly the enrichment of Akkermansia and Bifidobacterium, has emerged as a novel biomarker predictive of immunotherapy efficacy [178].

Beyond immunotherapy, the molecular profile of melanoma also informs the application of targeted therapies. The presence of a *BRAF* V600 mutation remains a pivotal predictive biomarker for targeted treatment, with combined BRAF and MEK inhibitor therapy yielding objective response rates of approximately 70% [179]. Additionally, mutations in *NRAS* have been associated with poorer prognoses, characterized by increased tumor aggressiveness, higher metastatic potential, and reduced melanoma-specific survival [180].

However, no single biomarker is sufficient; thus, integrative multi-biomarker approaches (combining TMB, gene expression signatures, and immune contexture) are being explored to improve predictive accuracy [178]. This multifaceted strategy holds promise for better personalizing therapy and prognostication in malignant melanoma.

Table 4 outlines the future directions discussed in this section. While many of these strategies remain investigational, they reflect the dynamic and rapidly progressing field of melanoma research and offer a foundation for future therapeutic innovations.

## Figures and Tables

**Figure 1 ijms-26-05370-f001:**
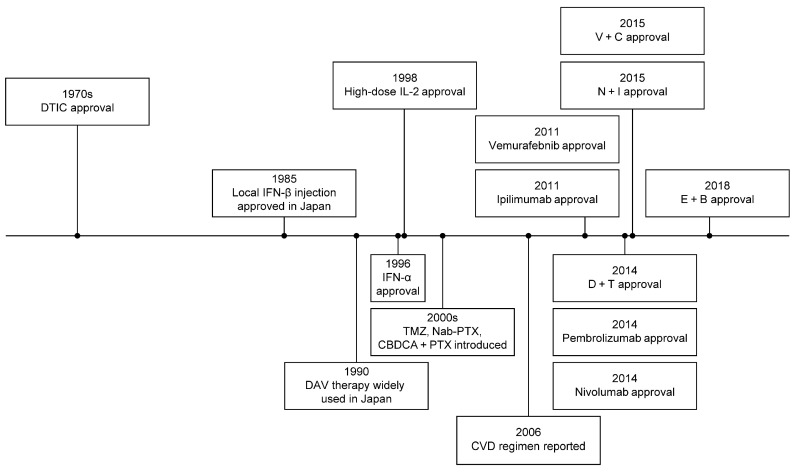
Historical development of melanoma treatment.

**Figure 2 ijms-26-05370-f002:**
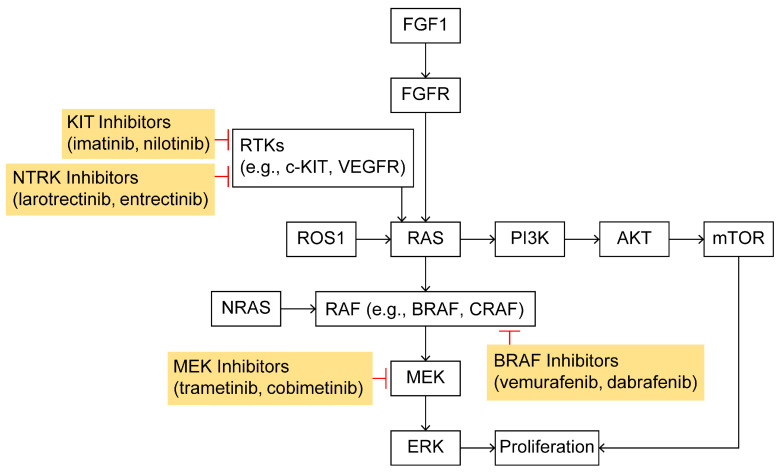
Pathogenic mechanisms and therapeutic targets in melanoma: the figure illustrates the RAS–RAF–MEK–MAPK signaling pathway activated by growth factor receptors, highlighting how specific genetic mutations contribute to melanoma development and how these mutations serve as therapeutic targets for currently available treatments.

**Table 1 ijms-26-05370-t001:** Summary of historical therapeutic approaches in melanoma and associated response rates [8,9,10,11,12,13,14,15,16,17,18,19,20,21,22,23,24,25,26,27,28,29,30,31,32].

Treatment Approach	Protocol	Efficacy	Key Considerations
Chemotherapy	Dacarbazine (DTIC)	ORR: 5–12%	Limited efficacy; standard before targeted therapy
	Temozolomide	ORR: 13.4%	No significant survival benefit over DTIC
	nab-Paclitaxel	ORR: 15%	No significant difference in OS compared to DTIC
	CBDCA + PTX (Carboplatin + Paclitaxel)	ORR: 18%	No significant difference from DTIC
	CVD (Cisplatin + Vindesine + DTIC)	ORR: 20–35%	No OS benefit over DTIC monotherapy
Cytokine Therapy	IL-2 (High-Dose Interleukin-2)	ORR: 15–20%; combined with SBRT: 54%	Durable response possible, but toxicity is high
	IFN-α (Interferon-Alpha)	ORR: low; no OS benefit	Used as adjuvant therapy but limited in recent use
	IFN-β (Interferon-Beta, Intralesional)	Complete response in case reports	Effective for in-transit/cutaneous metastases

Note: ORR: Objective Response Rate; OS: Overall Survival; SBRT: Stereotactic Body Radiation Therapy.

**Table 2 ijms-26-05370-t002:** Summary of clinical trials evaluating immune checkpoint inhibitors (ICIs) in advanced melanoma (non-adjuvant settings).

Year	Phase	Trial Name	Intervention vs. Control	Patient Population	ORR (%)	Reference
2010	III	MDX010-20	Ipilimumab + gp100 vs. gp100	Advanced, previously treated	10.9 vs. 5.7	[5]
2011	III	CA184-024	Ipilimumab + Dacarbazine vs. Dacarbazine	Treatment-naïve advanced	15.2 vs. 10.3	[89]
2014	III	CheckMate 066	Nivolumab vs. Dacarbazine	Treatment-naïve advanced *BRAF* WT	40.0 vs. 13.9	[96]
2014	III	CheckMate 037	Nivolumab vs. Chemotherapy	Advanced after ipilimumab or BRAF inhibitor	31.7 vs. 10.6	[6]
2014	II	KEYNOTE-002	Pembrolizumab vs. Chemotherapy	Advanced refractory to ipilimumab	25.0 vs. 4.0	[95]
2015	III	KEYNOTE-006	Pembrolizumab vs. Ipilimumab	Advanced (≤1 prior therapy)	33.7 vs. 11.9	[94]
2015	III	CheckMate 067	Nivolumab + Ipilimumab vs. Nivolumab vs. Ipilimumab	Treatment-naïve advanced	58 combo, 45 Nivo, 19 Ipi	[97,98]
2022	II/III	RELATIVITY-047	Nivolumab + Relatlimab vs. Nivolumab	Treatment-naïve advanced	43.1 vs. 32.6	[105]

Note: gp 100: glycoprotein 100; WT: wild-type; Nivo: nivolumab; Ipi: ipilimumab.

**Table 4 ijms-26-05370-t004:** Feature perspectives for melanoma treatment.

Therapeutic Strategy	Description
Inhibition of proliferative signaling pathways	Targeting constitutively active mutations (e.g., *BRAF*, *KIT*, *NRAS*, *TERT*, *CCND1*) to suppress melanoma proliferation; *TERT* and *CCND1* alterations are notable in acral melanoma.
Adoptive cell therapy using TILs	Lifileucel, an FDA-approved TIL therapy, shows efficacy after ICI and BRAF/MEK failure; limited by complexity, cost, and toxicity.
Melanoma antigen-targeted CAR-T therapy	TYP1-targeted CAR-T cells demonstrate promising preclinical efficacy; clinical translation in melanoma is ongoing.
Combining immune checkpoint inhibitors (ICIs) with other modalities	Strategies include dual checkpoint blockade, ICI with radiotherapy, oncolytic viruses, cancer vaccines, and TIL-ICI combinations.
Natural killer (NK) cell-based approaches	AR blockade may restore NK cell cytotoxicity and enhance ICI efficacy; under preclinical investigation.
Identification and validation of biomarkers	Biomarkers such as PD-L1, TMB, IFN-γ signature, TILs, LAG-3, HLA genotypes, and gut microbiota are critical for predicting treatment response.

Note: TIL: Tumor-Infiltrating Lymphocyte Therapy; CAR-T: Chimeric Antigen Receptor T-cell Therapy; Tumor Mutational Burden; Interferon; Lymphocyte Activation Gene-3.

## Data Availability

Not applicable.

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
