# Peer review of "Recent Advances in Molecular Research and Treatment for Melanoma in Asian Populations"

_ijms, 2025, doi:10.3390/ijms26115370_

Round 1
Reviewer 1 Report
Comments and Suggestions for Authors
The topic of this review is intriguing, however, there are different concerns:
1. It provides a chronological overview of melanoma treatments, transitioning from chemotherapy to targeted and immunotherapy approaches. However, in the whole manuscript, The shift from one sentence to another is abrupt. The transitions and the language should be more fluid.
2. The text moves logically from the limited effectiveness of chemotherapy to the breakthroughs of targeted and immune therapies. However, in some points, the phrases are confusing and should be restructured for clarity. In a lot of cases, there is a need for rewording.
3. Important FDA approvals and landmark therapies are appropriately mentioned. In some cases, the references are lacking (e.g anti-CTLA antibody ipilimumab was approved in 2011. please, add a citation).
4. In some points the review is too repetitive. It can be streamlined to avoid multiple mentions of “first” and “approved by FDA (e.g The first BRAF inhibitor, vemurafenib was approved by FDA in 2011," and "the first combination of BRAF and MEK was approved by FDA for the first time in 2015 (vemurafenib+cobimetinib)
5. Please pay attention to the verbal forms. Some sentences mix past and present tense.
6. The introduction section ends with a summary of the review but could benefit from a clearer thesis. What is the goal? What does this review highlight?
7. Figure 1 is appreciated, but please write a legend to write : (e.g T stands for, S stands for)
8. line 358, please introduce recent data concerning the expression of PD1 also on melanoma cells ( doi: 10.1038/s41419-025-07350-4; https://doi.org/10.1016/j.cell.2015.08.052) and comment on this expression. Can it be an advantage or a disadvantage for melanoma cells? It can be an intracellular or intercellular binding PDL1/PD-1?
9. Related to Immune Checkpoint Inhibitor Therapy, please add a table resuming the clinical trials.
10. In the context of Future Perspectives on Melanoma Management and the table I, the authors should mention recent data concerning melanoma escape from NK cells.
11. In addition, emerging data show that there are disparities based on sex in melanoma aggressiveness. Androgen antagonists improve the targeted therapy response; androgen receptor blocking is involved in senescence while its expression/activation is involved in migration and NK immune escape. In males, the prognosis is worse than in women. Androgens promote the shedding of some immune-check points, and the same increase is observed in patients with IV grade of melanoma (treated with pembrolizumab). What about these considerations? Can they open a new scenario?
The topic of this review is intriguing. However, to be more interesting and different from the others already present in literature, it should be more innovative. These are some hints.
Comments on the Quality of English Language
I have detailed it.
Author Response
Reviewer(s)' Comments to Author:
Reviewer: 1
Comments to the Author
The topic of this review is intriguing, however, there are different concerns:
- It provides a chronological overview of melanoma treatments, transitioning from chemotherapy to targeted and immunotherapy approaches. However, in the whole manuscript, The shift from one sentence to another is abrupt. The transitions and the language should be more fluid.
Response:
We thank the reviewer for pointing out this issue. In response, we have submitted the manuscript for professional English editing to improve the fluidity of transitions and overall language quality.
Editing was carried out by the service recommended by the journal: https://www.mdpi.com/authors/english.
- The text moves logically from the limited effectiveness of chemotherapy to the breakthroughs of targeted and immune therapies. However, in some points, the phrases are confusing and should be restructured for clarity. In a lot of cases, there is a need for rewording.
Response:
We thank the reviewer for pointing out this issue. To address the concerns regarding clarity and phrasing, we have submitted the manuscript for professional English editing, using the service recommended by the journal: https://www.mdpi.com/authors/english.
We believe that the revisions have significantly improved the clarity and readability of the manuscript.
- Important FDA approvals and landmark therapies are appropriately mentioned. In some cases, the references are lacking (e.g anti-CTLA antibody ipilimumab was approved in 2011. please, add a citation).
Response:
We thank the reviewer for pointing out this issue. As suggested, we have added an appropriate reference (Reference 5) to support the statement (3.3 Immunotherapy).
Additionally, we corrected a factual inaccuracy regarding the first FDA-approved combination of BRAF and MEK inhibitors. In the original version, we mistakenly indicated that vemurafenib plus cobimetinib was the first combination approved in 2015. This has been revised to reflect the fact that dabrafenib plus trametinib was the first combination approved in 2014.
- In some points the review is too repetitive. It can be streamlined to avoid multiple mentions of “first” and “approved by FDA (e.g The first BRAF inhibitor, vemurafenib was approved by FDA in 2011,"and "the first combination of BRAF and MEK was approved by FDA for the first time in 2015 (vemurafenib+cobimetinib)
 Response:
We thank the reviewer for pointing out this important issue.
In response, we have submitted the manuscript for professional English editing, using the service recommended by the journal: https://www.mdpi.com/authors/english.
We have revised the text to eliminate unnecessary repetitions and improve the overall flow and clarity.
- Please pay attention to the verbal forms. Some sentences mix past and present tense.
Response:
We thank the reviewer for pointing out this issue. To address the concern regarding inconsistent verb tenses, we have submitted the manuscript to the service recommended by the journal for professional English editing: https://www.mdpi.com/authors/english.
We believe that the revisions have corrected these inconsistencies and improved the overall quality of the manuscript.
- The introduction section ends with a summary of the review but could benefit from a clearer thesis. What is the goal? What does this review highlight?
  Response:
We thank the reviewer for pointing out this issue. In response, we have revised the final paragraph of the Introduction to better articulate the primary aim of this review. Specifically, we have emphasized the differences in melanoma characteristics and treatment responses between Asian and Caucasian populations, highlighting the importance of accumulating region-specific evidence to guide optimal management strategies in Asian patients. We have revised the manuscript as follows: “Importantly, we focus on the unique characteristics and clinical features of melanoma in Asian populations, which often differ from those observed in Caucasian patients. By highlighting these distinctions, this review aims to underscore the need for region-specific clinical evidence and to contribute to the development of more tailored treatment strategies for Asian patients” (Introduction).
- Figure 1 is appreciated, but please write a legend to write : (e.g T stands for, S stands for)
Response:
We thank the reviewer for pointing out this issue. We have revised Figure 1 to improve its clarity and have added a legend as suggested, specifying the meaning of each abbreviation.
- line 358, please introduce recent data concerning the expression of PD1 also on melanoma cells ( doi: 10.1038/s41419-025-07350-4; https://doi.org/10.1016/j.cell.2015.08.052) and comment on this expression. Can it be an advantage or a disadvantage for melanoma cells? It can be an intracellular or intercellular binding PDL1/PD-1?
Response:
We thank the reviewer for pointing out this issue. In response to your suggestion, we have revised the manuscript to incorporate recent findings regarding the expression of PD-1 on melanoma cells, as reported in the cited studies (DOI: 10.1038/s41419-025-07350-4; https://doi.org/10.1016/j.cell.2015.08.052). Specifically, we have addressed the potential functional implications of PD-1 expression on tumor cells. As suggested, we have described how PD-1 expression on melanoma cells may contribute to tumor progression. The interaction between PD-1 expressed on melanoma cells and PD-L1 on adjacent tumor or stromal cells has been shown to activate the mTOR signaling pathway, promoting tumor growth independently of immune cell involvement. Thus, PD-1 expression on tumor cells may serve as an advantage for melanoma proliferation. Conversely, this tumor-intrinsic PD-1 expression may render the cells more susceptible to PD-1 blockade, which could be considered a disadvantage from the tumor’s perspective.
We have added the following sentences to the revised manuscript: “It remains unclear precisely which patients with melanoma benefit from ICIs. PD-1 inhibitors, such as pembrolizumab and nivolumab, exert therapeutic effects primarily by blocking the interaction between the PD-1 receptor on T cells and its ligand, PD-L1, expressed on tumor cells. Initially, tumor PD-L1 expression was anticipated to correlate positively with clinical response. However, clinical implementation of PD-L1 as a reliable biomarker has been challenging due to its dynamic variability and intratumoral heterogeneity, as well as the observation that some patients with PD-L1-negative tumors also respond favorably to PD-1 inhibitor. Recent studies have demonstrated that PD-1, traditionally considered exclusive to immune cells, is also expressed on melanoma cells. In melanoma, the binding of PD-1 on tumor cells with PD-L1 expressed by surrounding tumor or stromal cells can activate the mTOR signaling pathway, thereby directly promoting tumor growth independently of immune cell interactions. Notably, PD-1 inhibitors may exert their therapeutic efficacy not only through immune modulation but also by directly interacting with PD-1 expressed on melanoma cell. Furthermore, recent evidence indicates that androgen receptor activation in melanoma cells may interfere with the efficacy of ICIs. Detailed biomarkers for treatment response will be discussed in a separate section; however, numerous studies are currently underway to identify reliable biomarkers predictive of response to ICIs” (3.3 Immunotherapy).
- Related to Immune Checkpoint Inhibitor Therapy, please add a table resuming the clinical trials.
Response:
We thank the reviewer for pointing out this issue. In response, we have created a new table (Table 2 in the revised manuscript) that summarizes the major clinical trials on immune checkpoint inhibitors in melanoma. We believe that this tabular overview will help readers by providing an accessible summary of the clinical evidence supporting immune checkpoint inhibitors, complementing the descriptive text and addressing the reviewer’s request.
- In the context of Future Perspectives on Melanoma Management and the table I, the authors should mention recent data concerning melanoma escape from NK cells.
Response:
We thank the reviewer for the insightful suggestion. In response, we have revised Table 4 to include recent data concerning melanoma escape from natural killer (NK) cells. This addition aligns with the newly included paragraph under “Future Perspectives on Melanoma Management”, which discusses the mechanistic and therapeutic implications of AR signaling in NK cell regulation.
- In addition, emerging data show that there are disparities based on sex in melanoma aggressiveness. Androgen antagonists improve the targeted therapy response; androgen receptor blocking is involved in senescence while its expression/activation is involved in migration and NK immune escape. In males, the prognosis is worse than in women. Androgens promote the shedding of some immune-check points, and the same increase is observed in patients with IV grade of melanoma (treated with pembrolizumab). What about these considerations? Can they open a new scenario?
The topic of this review is intriguing. However, to be more interesting and different from the others already present in literature, it should be more innovative. These are some hints.
Response:
We thank the reviewer for raising this important issue and for their insightful suggestions.
We fully agree that this represents a novel and underexplored dimension of melanoma biology that deserves further attention.
In response to this valuable suggestion, we have revised the section titled “Future Perspectives on Melanoma Management” to include a new paragraph (the 14th paragraph under point 5) discussing the impact of AR signaling on NK cell-mediated immunity and immune evasion. Specifically, we elaborate on how ligand-activated AR promotes the shedding of the NKG2D ligand MICA via the upregulation of ADAM10, leading to reduced NK cell recognition and immune escape in melanoma. This mechanism is particularly relevant in male patients, who exhibit lower NK cell activity compared to females, potentially contributing to sex-based differences in clinical outcomes.
Furthermore, we discuss the therapeutic potential of AR antagonists, such as bicalutamide and enzalutamide, in restoring NK-mediated cytotoxicity and enhancing the efficacy of immune checkpoint inhibitors. These findings may indeed open a promising new avenue for melanoma treatment, especially in male patients or those with high AR expression, as suggested by recent preclinical studies.
We believe that the inclusion of this topic adds a novel and innovative perspective to our review and aligns with the reviewer’s thoughtful suggestion to distinguish our work from the existing literature.

Reviewer 2 Report
Comments and Suggestions for Authors
Kado and Komine presented a manuscript entitled “Recent advances in molecular research on melanoma therapy”.
The topic is well described; however, some questions must be resolved before publication:
1)The authors should mention the figures and the table in the main text.
Figure 1: References should be added to the figure
Figure 2: The authors should better explain what they want to represent in this figure. The figure is too simplified, losing the main message.
2) To be clearer, I suggest adding to the title that this review focuses on the Asian population.
Author Response
Comments to the Author
Kado and Komine presented a manuscript entitled “Recent advances in molecular research on melanoma therapy”.
The topic is well described; however, some questions must be resolved before publication:
- The authors should mention the figures and the table in the main text.
Figure 1: References should be added to the figure
Figure 2: The authors should better explain what they want to represent in this figure. The figure is too simplified, losing the main message.
Response:
We thank the reviewer for pointing out these important issues and suggestions.
We have revised the main text to appropriately mention the figures and the table.
Regarding Figure 1, as it is an original illustration, we have not included any references within the figure itself. We appreciate your understanding and kind consideration.
For Figure 2, we have revised the figure to better convey the intended message and have provided a more detailed explanation in the legend.
- To be clearer, I suggest adding to the title that this review focuses on the Asian
population.
Response:
Thank you very much for your valuable suggestion.
We have revised the title as recommended:
“Recent Advances in Molecular Research and Treatment for Melanoma in Asian Populations”

Reviewer 3 Report
Comments and Suggestions for Authors
The manuscript "Recent advances in molecular research on melanoma therapy" summarizes the treatments available for melanoma, issues with these treatments, followed by future perspective. The information provided in comprehensive but not recent. Most of the citations are not recent. The citations are mostly from 2010-2018- which cannot be considered as recent.
After introduction, the authors described the historical development which may be in the form of a table or chart. This is followed by current advances in the treatment- again in this section all citations till 41 are very old. Only 42 and 43 are recent and after that again are from 2014-2018. This should be summarized as a table and using a figure.
Section 3 and 4 are ok and the focus should be on recent advances in immunotherapy. Why immunotherapy is not efficient and effective followed by what should be done to improve treatment.
Comments on the Quality of English LanguageThe English language is ok.
Author Response
Comments to the Author
The manuscript "Recent advances in molecular research on melanoma therapy" summarizes the treatments available for melanoma, issues with these treatments, followed by future perspective. The information provided in comprehensive but not recent. Most of the citations are not recent. The citations are mostly from 2010-2018- which cannot be considered as recent.
Response:
We thank the reviewer for pointing out this important issue.
As suggested, we recognized that many of the references were outdated and have revised the manuscript accordingly.
Specifically, we have updated the “Future Perspectives” section by including more recent references, incorporating recent clinical trial data, emerging research on the role of natural killer (NK) cells, and the latest findings related to biomarkers.
After introduction, the authors described the historical development which may be in the form of a table or chart. This is followed by current advances in the treatment- again in this section all citations till 41 are very old. Only 42 and 43 are recent and after that again are from 2014-2018. This should be summarized as a table and using a figure.
Response:
We thank the reviewer for this helpful suggestion. Accordingly, we have organized the historical development of chemotherapy into Table 1 and summarized the current advances in melanoma treatment in Table 2 to improve clarity and readability.
Section 3 and 4 are ok and the focus should be on recent advances in immunotherapy. Why immunotherapy is not efficient and effective followed by what should be done to improve treatment.
Response:
We appreciate the reviewer’s insightful comment.
To address this, we have considered the combination of immune checkpoint inhibitors with other therapeutic approaches as a potential solution to improve efficacy.
In the subsection “Future Perspectives on Melanoma Management”, we have incorporated new content summarizing recent clinical trials on the combination of immune checkpoint inhibitors with radiotherapy, chemotherapy, oncolytic viruses, therapeutic cancer vaccines, and tumor-infiltrating lymphocyte (TIL) therapy, along with relevant updated references.
Furthermore, we have summarized these clinical trials in a newly added Table 3 for clarity.

Reviewer 4 Report
Comments and Suggestions for Authors
There are some comments.
1. It is recommended to revise the title to:
"Recent Advances in Molecular Research and Treatment for Melanoma"
2. It would be better to add a section discussing the molecular classification of melanoma, such as the TCGA classification, to provide a comprehensive overview of the genetic subtypes and their clinical relevance.
3. Including a subsection on molecular biomarkers associated with prognosis and treatment response is advised.
4. Please revise Table 2 by the International Journal of Molecular Sciences (IJMS) formatting guidelines.
5. Ensure that the words Figures 1 and 2 are correctly inserted into the main text and appropriately referenced within the manuscript.
6. A list of abbreviations used throughout the manuscript would be better.
7. Revise all references to comply with the IJMS citation and formatting style.
Comments on the Quality of English LanguagePlease check English grammar and spelling.
Please check the gene name in Italics.
Author Response
There are some comments.
1. It is recommended to revise the title to:
"Recent Advances in Molecular Research and Treatment for Melanoma"
Response:
Thank you very much for your valuable suggestion.
We have revised the title as recommended. Additionally, considering Reviewer 2’s comment requesting emphasis on the Asian population, we have further modified the title to “Recent Advances in Molecular Research and Treatment for Melanoma in Asian Populations”.
- It would be better to add a section discussing the molecular classification of melanoma, such as the TCGA classification, to provide a comprehensive overview of the genetic subtypes and their clinical relevance.
Response:
We thank the reviewer for the valuable comment. In response, we have added a new subsection titled “3.1 Molecular Classification of Melanoma (The Cancer Genome Atlas Project)”, in which we provide an overview of the TCGA classification, focusing on the major genetic subtypes and their clinical relevance.
- Including a subsection on molecular biomarkers associated with prognosis and treatment response is advised.
Response:
Thank you very much for your valuable suggestion. In response, we have added a new subsection under “Future Perspectives on Melanoma Management” that focuses on molecular biomarkers associated with prognosis and treatment responses. This new section discusses key biomarkers, including PD-L1 expression, tumor mutational burden (TMB), IFN-γ gene expression signatures, tumor-infiltrating lymphocytes (TILs), immunosuppressive cell populations, IDO1, LAG-3, HLA genotypes, and gut microbiome composition. We also emphasize the importance of integrative multi-biomarker approaches to enhance predictive accuracy and guide personalized treatment strategies. We believe that this addition strengthens the overall scope of the manuscript and aligns well with the reviewer’s suggestion.
- Please revise Table 2 by the International Journal of Molecular Sciences (IJMS) formatting guidelines.
Response:
We thank the reviewer for this helpful comment.
During the revision process, we created Tables 1–4 and formatted all the tables, including Table 2, according to the International Journal of Molecular Sciences (IJMS) formatting guidelines.
- Ensure that the words Figures 1 and 2 are correctly inserted into the main text and appropriately referenced within the manuscript.
Response:
We thank the reviewer for pointing out this issue.
As suggested, we have revised the manuscript to ensure that Figures 1 and 2 are correctly inserted and appropriately referenced in the main text.
- A list of abbreviations used throughout the manuscript would be better.
Response:
We thank the reviewer for this helpful suggestion.
We have addressed this issue appropriately by creating a list of abbreviations used throughout the manuscript.
These revisions also underwent professional English editing by the service recommended by the journal: https://www.mdpi.com/authors/english.
- Revise all references to comply with the IJMS citation and formatting style.
Comments on the Quality of English Language
Please check English grammar and spelling.
Please check the gene name in Italics.
Response:
We thank the reviewer for their helpful comments.
We have revised all the references to comply with the IJMS citation and formatting style.
Additionally, we have checked and corrected the English grammar, spelling, and gene name formatting (in italics), using the professional English editing service recommended by the journal: https://www.mdpi.com/authors/english.

Round 2
Reviewer 1 Report
Comments and Suggestions for Authors
It can be accepted
Author Response
Reviewer(s)' Comments to Author:
Reviewer: 1
Comments to the Author
It can be accepted
Response:
Thank you very much for your positive evaluation of our revised manuscript. We greatly appreciate your support and constructive feedback throughout the review process.

Reviewer 3 Report
Comments and Suggestions for Authors
None
Author Response
Reviewer: 3
Comments to the Author
None
Response:
Thank you for taking the time to review our manuscript. We are grateful for your consideration and have ensured that all aspects of the manuscript maintain academic clarity and accuracy.

Reviewer 4 Report
Comments and Suggestions for Authors
The manuscript was well revised.
There are some comments.
There are two minor comments.
1. The abbreviations used in Tables 1, 2 and 4 should be clearly defined in the footnotes of those tables, respectively.
2. Please double-check the references cited in Table 3. For example, 'NCT03340129' appears to be a clinical trial ID rather than a reference number.
Comments on the Quality of English LanguagePlease check English grammar and spelling
Author Response
Reviewer: 4
Comments to the Author
The manuscript was well revised.
There are some comments.
There are two minor comments.
- The abbreviations used in Tables 1, 2 and 4 should be clearly defined in the footnotes of those tables, respectively.
Response:
Thank you very much for your thoughtful suggestion. We have carefully reviewed all the abbreviations used in Tables 1, 2, and 4 and have now clearly defined each of them in the corresponding footnotes of the tables. We believe that this revision improves the clarity and readability of the data presented.
- Please double-check the references cited in Table 3. For example, 'NCT03340129' appears to be a clinical trial ID rather than a reference number.
Response:
Thank you for pointing this out. We have rechecked all the references cited in Table 3 and have corrected the format of the clinical trial identifiers accordingly. Specifically, ‘NCT03340129’ and other clinical trial IDs have been properly cited as clinical trials with corresponding information included in the reference list, following the journal's guidelines.
Comments on the Quality of English Language
Please check English grammar and spelling
Response:
Thank you for your comment regarding the quality of the English language. In response, we have requested a comprehensive re-editing of the manuscript by the MDPI English editing service, as previously recommended. The entire manuscript has been carefully reviewed and revised to improve grammar, spelling, and overall linguistic clarity. We believe that the revised version now meets the standards of academic English suitable for publication.
